# Extreme positive epistasis for fitness in monosomic yeast strains

Hanna Tutaj[1]*, Katarzyna Tomala[1], Adrian Pirog[1], Marzena Marszałek[1,2], Ryszard Korona[1]*

[1]Institute of Environmental Sciences, Faculty of Biology, Jagiellonian University, Cracow, Poland; [2]Doctoral School of Exact and Natural Sciences, Jagiellonian University, Cracow, Poland

## eLife Assessment

This study offers **important** insights into the generation and maintenance of monosomic yeast lines and is, to our knowledge, the first to evaluate gene expression in yeast monosomies. The research introduces an innovative method to assess epistasis between genes on the same chromosome, providing **solid** evidence for positive epistatic interactions affecting fitness. Although the authors have substantially improved the methodology and interpretation during revision, questions regarding the interpretation of the transcriptome data have not been completely addressed.

**\*For correspondence:**
hanna.tutaj@doctoral.uj.edu.pl (HT);
ryszard.korona@uj.edu.pl (RK)

**Competing interest:** The authors declare that no competing interests exist.

**Abstract** The loss of a single chromosome in a diploid organism halves the dosage of many genes and is usually accompanied by a substantial decrease in fitness. We asked whether this decrease simply reflects the joint damage caused by individual gene dosage deficiencies. We measured the fitness effects of single heterozygous gene deletions in yeast and combined them for each chromosome. This predicted a negative growth rate, that is, lethality, for multiple monosomies. However, monosomic strains remained alive and grew as if much (often most) of the damage caused by single mutations had disappeared, revealing an exceptionally large and positive epistatic component of fitness. We looked for functional explanations by analyzing the transcriptomes. There was no evidence of increased (compensatory) gene expression on the monosomic chromosomes. Nor were there signs of the cellular stress response that would be expected if monosomy led to protein destabilization and thus cytotoxicity. Instead, all monosomic strains showed extensive upregulation of genes encoding ribosomal proteins, but in an indiscriminate manner that did not correspond to their altered dosage. This response did not restore the stoichiometry required for efficient biosynthesis, which probably became growth limiting, making all other mutation-induced metabolic defects much less important. In general, the modular structure of the cell leads to an effective fragmentation of the total mutational load. Defects outside the module(s) currently defining fitness lose at least some of their relevance, producing the epiphenomenon of positive interactions between individually negative effects.

## Introduction

Cells and entire organisms can tolerate a surprisingly large number of deleterious mutations, especially when they affect gene function partially rather than completely, for example, when only one of two alleles is inactivated. In fact, a typical human germline cell has as many as about 100 inactive genes, usually complemented by active alleles (*MacArthur et al., 2012*). Somatic cell lines in healthy tissues can add dozens of new (nearly always heterozygous) mutations every year (*Moore et al., 2021*). Understanding how this high burden of DNA damage translates into a relatively low mutational

load, defined as a reduction in fitness, has proven to be a real challenge (*Henn et al., 2015*). One possible explanation is that mutations typically have negligible effects, especially when heterozygous. Another is that deleterious mutations are balanced by compensatory mutations, although the latter may not yet have been identified. However, it is also possible that the negative effects of mutations are significantly reduced when they are numerous, that is, they effectively offset each other through positive epistasis.

Because the fitness architecture of complex organisms is particularly difficult to disentangle, researchers have turned to simpler systems. In the model organism *Saccharomyces cerevisiae*, collections of engineered single-gene mutations, such as complete deletions, have been created and used to estimate selection and dominance coefficients, as well as the effects of interactions between a few (typically two) loci (*Costanzo et al., 2010*; *Giaever et al., 2002*; *Steinmetz et al., 2002*). It would be much more difficult to combine large numbers of deleterious mutations. This inevitably long process would expose the strains under construction to a high probability of developing fitness-restoring genetic adaptations, which are known to occur even with limited cell proliferation (*Szamecz et al., 2014*). A rapid way to generate genomes with extensive loads of known mutations is needed. With this in mind, we noted that the budding yeast genome consists of approximately 5000 genes, distributed in a largely random manner across 16 chromosomes. We reasoned that monosomy, the loss of one chromosome in an otherwise regular diploid (2*n*-1), would mimic the sudden mutation (partial inactivation) of numerous and randomly assembled genes.

## Results

### Viability of monosomic strains

We began our study by inserting three markers into one chromosome within each homologous pair. These were two *URA3* genes, each at the center of every arm, and one copy of *kan* near a centromere (*Tutaj et al., 2022*). Since its homolog was unmarked, mutants lacking the markers signaled a possible loss of the marked chromosome ('Materials and methods'). We expected that such generated true monosomic strains (M1, M2,..., M16) would show visible growth defects, which we indeed observed, except for M1 (*Figure 1A*). The loss of markers and slow growth were only hints, and often turned out to be false when DNA sequencing revealed that the expected loss of an entire single chromosome did not occur, or that it did occur but was accompanied by other changes. *Figure 1B* shows isolates that passed the sequencing test. As shown in *Figure 1C*, small colonies formed by monosomic cells were regularly invaded by much larger ones. For each monosomic, dozens of the latter were re-streaked and found to reproduce the fast-growing phenotype almost universally. To prevent the monosomics from overgrowing by the fast-growers, we had to keep the cultures small, down to 5 μL of YPD medium in the case of M5 or M15. The cultures had to be replicated (in tens or hundreds) so that a sufficient number of those dominated by the slow-growing cells could still be found and replenished.

The procedure described above could not be completed for three chromosomes—VII, XII, and XIII—although we paid special attention to them. Regarding the screens for M7 and M13, we did not find a single mutant that lacked the markers, grew slowly, and could rapidly revert to a normal phenotype. Instead, we regularly encountered isolates of the desired genotype with no growth defect and a diploid genome. Hypothetically, two mitotic crossovers within a pair of homologous chromosomes could produce an offspring cell without the markers. We were able to reject this possibility based on the results of the present study and a previous study (*Tutaj et al., 2022*). Relevant data and calculations are provided in Appendix 1. The best explanation is that the marked chromosomes VII and XIII were lost and their unmarked homologs underwent endoreduplication shortly thereafter. The endoreduplication could result from non-disjunction of a monosomic chromosome. Thus, we suggest that the cells of M7 and M13 were viable but suffered from very slow growth, high rate of endoreduplication, or both, and were thus especially rapidly overgrown by revertants. The search for M12 yielded unusually numerous and genotypically variable isolates, but none that were correct in terms of growth phenotype or DNA copy number mapping ('Materials and methods', Appendix 1). Chromosome XII contains the only yeast rDNA region and is a strong recombinational hotspot (*Tutaj et al., 2022*). Therefore, we suspended attempts to derive M12, especially since, as we show below, its predicted fitness loss from haploinsufficiency was well within the range covered by other chromosomes.

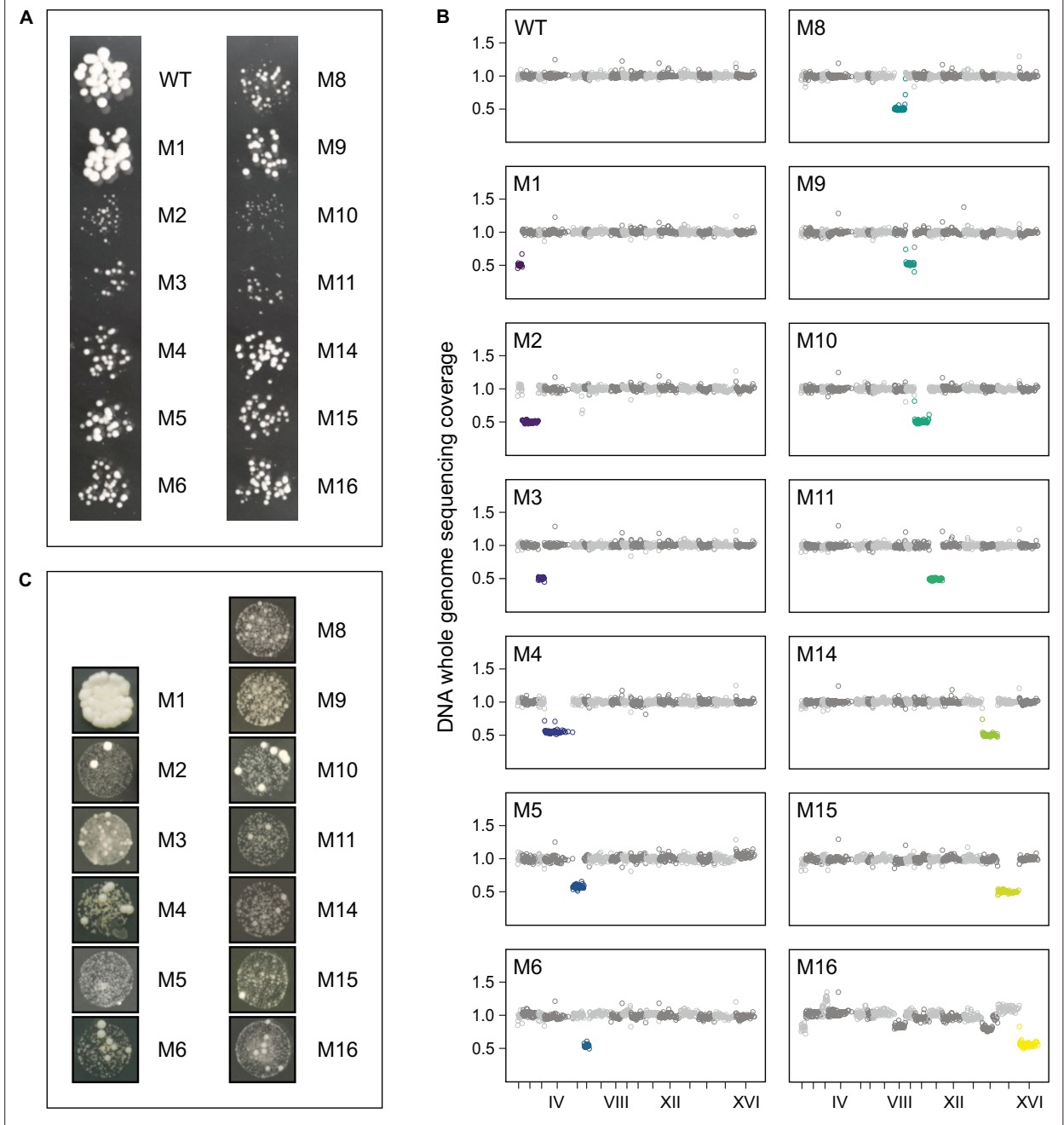

**Figure 1.** Diploid yeast strains lacking single chromosomes (monosomics). (**A**) Representative colonies of wild-type diploid (WT) and monosomic mutant strains (M1...M16) when grown together for 24 hr at 30°C on agar-solidified YPD medium. (**B**) DNA whole-genome sequencing coverage. (**C**) Examples of large colonies emerging among the small monosomic ones.

In the wild, monosomic isolates have only been found for a few and the shortest chromosomes (*Peter et al., 2018*). Similar examples of monosomy have been detected in mutation-accumulation experiments (*Sui et al., 2020*; *Zhu et al., 2014*). It is uncertain whether these monosomies were simply tolerated, compensated by other changes, or conditionally adaptive in certain environments. There have been laboratory experiments suggesting that all monosomies can be induced in yeast, though indicated only by loss of heterozygosity at selected loci, but they may have been transient (*Beach et al., 2017*; *Reid et al., 2008*). We suggest that past work on yeast monosomy should be taken with a grain of salt and that future work should be properly planned. Colonies of putative monosomics may actually be diploids or aneuploids. Even if they were initially truly monosomic, they could

easily have been overtaken by endoreduplications or other compensations for chromosome loss. This could happen even if a moderately sized colony was formed from an initially monosomic cell. Here, we combined the marker-based approach with genome-wide assays based on sequencing of DNA (reported above) and RNA (below) to confirm that nearly all yeast monosomics are viable.

## Epistasis for fitness

Before studying epistasis between multiple mutations, their individual effects must be known. With respect to yeast growth rate, virtually none of the single-gene deletions can increase it (*Sliwa and Korona, 2005*). About 1/5 of them stop growth altogether and another 1/5 slow it down detectably (*Giaever et al., 2002*). These estimates are based on homozygous or haploid strains under optimal laboratory conditions. Heterozygous deletions, again under good conditions, often show morphological changes, but their growth rate, that is, proliferative fitness, is rarely and only modestly affected (*Deutschbauer et al., 2005*; *Marek and Korona, 2016*; *Ohnuki and Ohya, 2018*). In 'Materials and

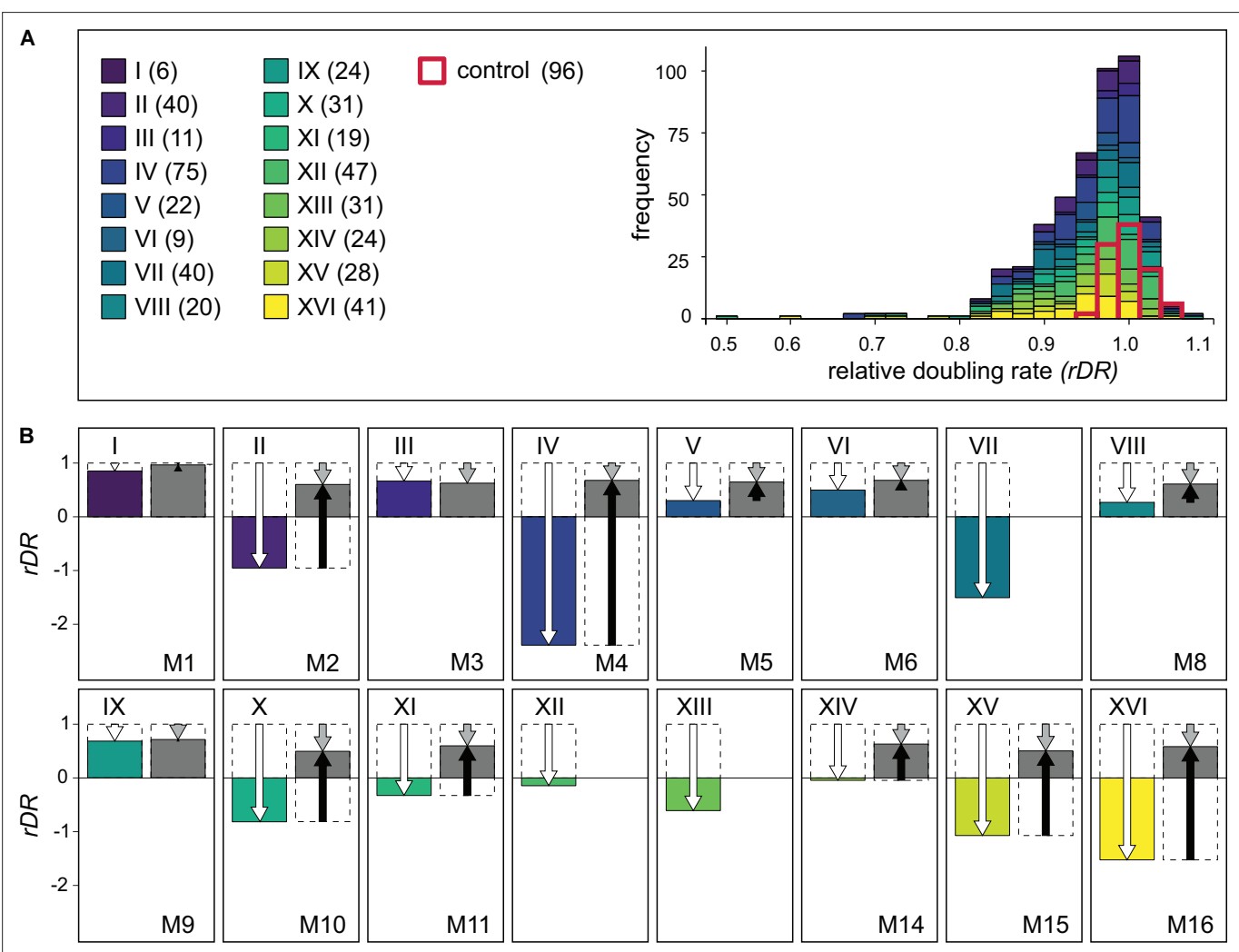

**Figure 2.** Contribution of positive epistasis to fitness of monosomics. (**A**) Growth performance of heterozygous single-gene deletion strains tested in this study. Left: list of chromosomes with numbers of assayed deletion strains. Right: frequency distribution of *rDR* (doubling rate related to that of the control). (**B**) Growth performance of monosomics. Colored bars represent expected performance calculated as a sum of the single-gene effects per chromosome, $rDR_E = 1 + \sum d_i$, where $d_i = rDR_i - 1$ (deviation from the control). Gray bars show the observed performance of monosomic strains, $rDR_M$. White arrows mark the expected departure from wild-type fitness (expected genetic load), gray the observed one. Black arrows show the extent and direction of epistasis. Three monosomics were not included in these assays (see the main text).

The online version of this article includes the following source data for figure 2:

**Source data 1.** Relative doubling rates of individual heterozygous deletion strains and monosomic strains related to *Figure 2A and B*.

methods', we explain how we selected 468 heterozygous single-deletion strains as potentially (haplo) insufficient for growth on the rich medium, YPD. We performed repeated growth rate measurements for single-deletion strains and compared them to a set of carefully selected control strains to make our estimates as accurate and reliable as possible. The doubling rates of individual heterozygous deletions were divided by that of the control to obtain relative doubling rates (*rDRs*) (*Figure 2—source data 1*). The frequency distribution of *rDR* is shown in *Figure 2A*. It shows that the negative effects of the deletions were undoubtedly present, although predictably small, a substantial fraction of them were in the range of bidirectional effects, most likely composed of phenotypic plasticity and measurement error. (The existence of phenotypic variation not attributable to gene deletions is revealed by the variation within a genetically homogeneous control.)

Epistasis is absent when an effect of the gene variant does not depend on the genetic content of other loci. In the case of fitness, a non-neutral mutation corresponds to a proportional change in wild-type fitness (*Crow and Kimura, 1970*). The absence of epistasis means that the fitness quotient of a given mutation remains unchanged regardless of whether it is the only one present in an individual or accompanied by other fitness-affecting mutations. Accepting this postulate is equivalent to adopting a multiplicative model of fitness structure, in which the fitness of a genotype involving $n$ loci is the product of $n$ respective quotients. The model is additive for log-fitness because such a transformation turns a quotient into a deviation from one and a product of quotients into a sum of such deviations. Since we are working with a ($\log_2$) transformed measure of fitness, the deviation caused by an $i$th gene deletion is equal to $d_i = rDR_i - 1$. The expected effect of multiple loci is the sum of all deviations involved, $rDR_E = 1 + \sum d_i$. (Derivation provided in Appendix 2.) To predict how each of the 16 monosomic strains should perform, we summed the deviations caused by the individual deletions present on each chromosome (obtaining 16 values of $rDR_E$). We summed both negative and positive values of $d$ to account for non-genetic variation of estimates. We then compared the predicted values of $rDR_E$ with the corresponding values of $rDR_M$, that is, estimates obtained experimentally for the actual monosomic strains (*Figure 2—source data 1*). *Figure 2B* shows the expected and obtained values of monosomics' *rDR*, where epistasis is equal to the difference between them. The epistasis between multiple deleterious gene deletions turned out to be not only positive but also large, much larger than that observed for just pairs of deletions (*Jasnos and Korona, 2007*). Most strikingly, some monosomic strains were expected to have a negative doubling rate, that is, were

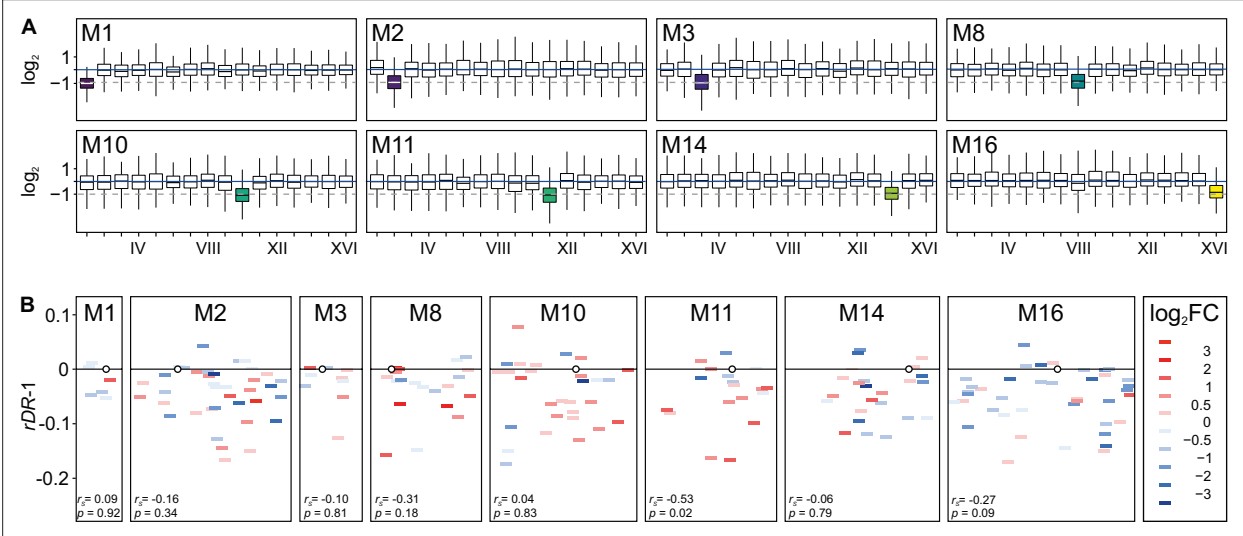

**Figure 3.** Absence of transcriptional compensation in monosomic strains. (**A**) Halved RNA production on monosomic chromosomes. For every open reading frame (ORF), the obtained number of RNA-seq reads was divided by the number expected for it under expression being constant over an entire genome. (**B**) Expression under monosomy vs. single-deletion fitness (*rDR*). X-axis shows the length of a monosomic chromosome with centromeres marked as circles and gene deletions as bars; colors show the effect of monosomy on the level of a particular mRNA with a particular color showing a range of $\log_2$ fold change (FC) relative to the control. Y-axis: the difference in *rDR* between a single-gene deletion strain and the control, $d = rDR - 1$. The correlation between fitness effect ($rDR - 1$) and shift in expression ($\log_2 FC$) is reported as Spearman's coefficient $r_s$ with associated and p-value (not corrected for the multiplicity of comparisons).

predicted to be effectively lethal, but turned out to be able to proliferate. For several monosomics, and especially M4, most of the predicted mutational load was canceled out by epistasis. The contribution of epistasis to fitness seemed so large that it might be difficult to accept without a functional rationalization.

## Transcriptome reaction

Epistasis can be considered in a purely abstract way as a deviation from additivity/multiplicity. But it must have a biological explanation. Our attempt to find it began with the isolation and quantitative analysis of mRNA from the ancestral diploid strain and eight monosomic strains. (The selection was random. It nevertheless resulted in the inclusion of strains in which the level of epistasis was low, M1 and M8, or high, M2 and M16.) The first question was obvious: was the decrease in gene dosage compensated by increased expression? *Figure 3A* shows that there was no detectable increase in the average intensity of transcription on the monosomic chromosomes. These averages depend on hundreds of genes and would therefore be largely insensitive to increased expression of important but relatively few genes. In this experiment, we knew which genes would be most rewarding to upregulate, those that were most haploinsufficient, and could focus our attention on them. *Figure 3B* shows that the fitness effect of a single-gene deletion did not correlate with the expression of this gene in a monosomic strain. Thus, neither physical underrepresentation (single copy) nor functional

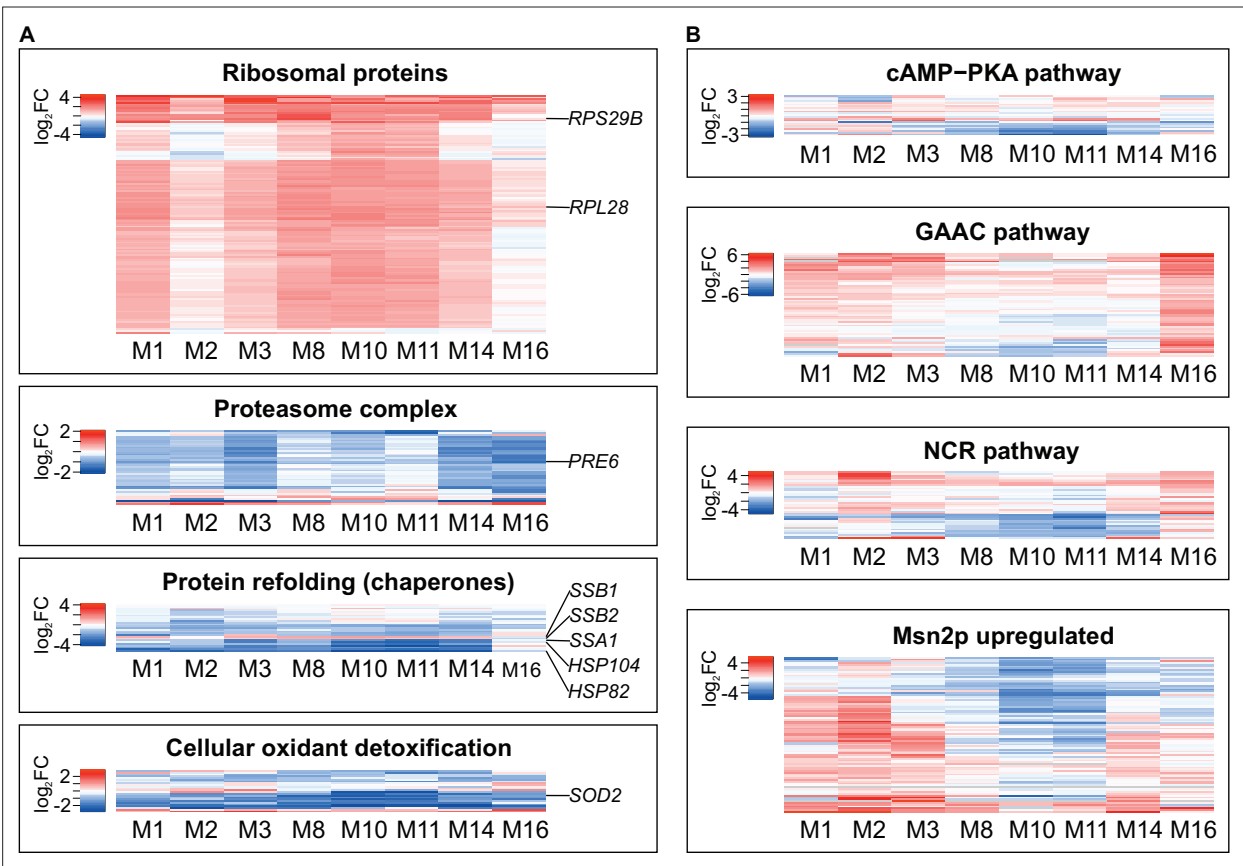

**Figure 4.** Parallel and divergent shifts in transcriptomes of monosomic strains. Heat maps show monosomic mRNA frequencies divided by respective diploid (control) ones. (**A**) Gene Ontology categories selected to demonstrate similarities in transcriptional profiles of monosomic strains. (**B**) Regulons demonstrating differences in gene expression between monosomic strains. Expanded versions of all panels can be found in *Figure 4—figure supplement 1*.

The online version of this article includes the following source data and figure supplement(s) for figure 4:

**Source data 1.** Differential gene expression in monosomic strains related to *Figures 3B and 4A, B*.

**Figure supplement 1.** Parallel and divergent shifts in transcriptomes of monosomic strains.

importance (impact on growth) triggered an increase in expression of the genes on monosomic chromosomes: their mRNA level was halved on average, although individual genes could show either up- or downshifts.

Monosomy could have altered the expression of any genes, not just those on the affected chromosomes. We aimed to find any distinct and functionally interpretable patterns of adjustment that might explain how the functioning of monosomic strains was perturbed. We found multiple statistically significant up- and downregulations in the transcriptomes of each monosomic strain analyzed (*Figure 4—source data 1*). Both parallelisms and incongruencies between them were visible.

*Figure 4A* shows the similarities. The translational apparatus of the cytosol was on average significantly and universally upregulated, as evidenced by transcripts encoding proteins that build both large (e.g., *RPL28*) and small (*RPS29B*) ribosomal subunits. Cytosolic proteolysis was downregulated (core proteasome component *PRE6*). In remarkable agreement, chaperones required to fold newly synthesized peptides were upregulated (*SSB1/2*), while those required to direct destabilized chains to degradation were downregulated (*SSA1, HSP82, HSP104*). Transcripts encoding mitochondrial proteins were generally less abundant, with a marked decrease in the expression of several genes encoding the electron transport chain (*Figure 4—source data 1*). In parallel, the expression of the antioxidant machinery was downregulated, especially that of a major ROS scavenger (*SOD2*).

To find transcriptomic differences, we examined groups of genes, each responding to a known signal. This increased the chances of detecting statistical significance and functional divergence. In our experiment, all strains received identical external signals. Therefore, different expression patterns would indicate different internal perturbations. Indeed, as shown in *Figure 4B*, substantial variation was detected within several regulons: cAMP-PKA (glucose-activated signaling), NCR (nitrogen catabolism repression), GAAC (general amino acid control), and Msn2up (activation by a broad range of stresses). These differences underscore the significance of the above reported uniformity in the pattern of biosynthetic upregulation and proteolytic downregulation.

All of the above considerations refer to the relative abundance of mRNA species. We also attempted to compare the absolute size of the transcriptome in wild-type BY and three monosomic strains: M1, M2, and M3. We added an admixture of specific external mRNA to provide a quantitative reference, or 'spike', to known total cell volumes of each strain and repeated our assays. The results are shown in *Figure 5*. In summary, the spike represented 4.04% of total mRNA for the BY sample and 6.08, 15.6, and 24.2% for the respective monosomic strains. Thus, monosomy would be associated with a decrease in the absolute level of mRNA. A possible caveat is that the monosomic cells had an altered morphology. They were more rounded in shape and approximately 60% larger in long axis, so that their individual volumes were several times larger than wild-type. Thus, the size of the transcriptome for M1 and M2 would not be reduced but actually increased if calculated per whole

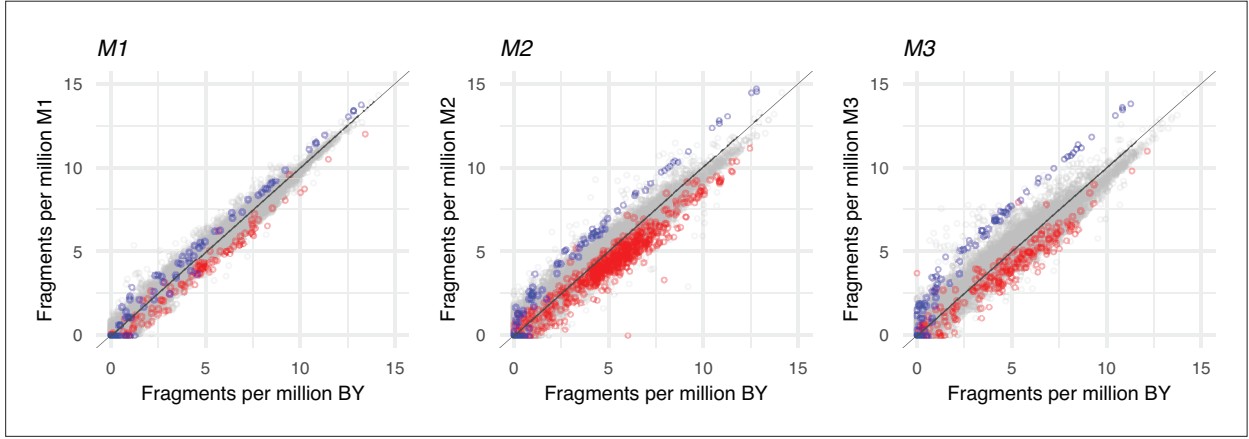

**Figure 5.** Correlation of counts of individual mRNAs between wild-type and monosomic strains. Counts are expressed as fractions of either wild-type BY or monosomic M1, M2, and M3 total transcriptomes. Gray circles represent mRNAs from the unaffected 15 chromosomes and group around the diagonal. Blue represent spike mRNAs. Red circles represent mRNAs from the monosomic chromosomes (I, II, or III in the respective graphs). Note that the monosomic counts are, as expected, underrepresented in the respective monosomic strains (red circles are below the diagonal). Monosomic counts of spike are higher than that of BY (blue circles are above the diagonal). As reported in the main text, the total fraction of spike counts in BY is 4.04%. Analogous sums for M1, M2, and M3 are 6.08, 15.6, and 24.2%. This can be seen here as an increasing distance between the gray and blue circles.

cell (one nucleus). Indeed, one explanation for the observed epistasis for viability could be an ample overproduction of all transcripts, so that even those halved by monosomy are sufficiently abundant. We consider the cell volume calculations more appropriate and therefore accept that the transcripts from monosomic chromosomes were even less abundant than half of the wild strain abundances. The apparent decrease in the cellular density of the transcriptome in the cytoplasm of monosomics is not paradoxical because the RNA content, including ribosomal protein mRNAs, decreases with slowing growth (*García-Martínez et al., 2016*; *Warner, 1999*). However, even if one accepts that monosomic transcriptomes were expanded (per nucleus), they would still be unbalanced. Thus, if the lack of stoichiometry between proteins and the resulting low efficiency of ribosome assembly, rather than the simple scarcity of specific elements, is the major consequence of monosomy, it would remain so regardless of the absolute size of the transcriptome.

## Discussion

The successful derivation and maintenance of monosomic strains demonstrated that it is possible to obtain an a priori designed set of strains with hundreds of partially inactive genes, dozens of which are known to have deleterious fitness effects. It implies that the ability to remain viable despite carrying extensive mutational burdens is not limited to specific combinations of genes, but would likely apply to a wide variety of them. Viability was supported by the epistatic effect, which was of positive value and remarkable strength, but may still be underestimated in our study. It was shown that many of the single deletions with negative fitness effects had undergone compensatory evolution under recurrent replication needed to maintain the strains carrying them (*Puddu et al., 2019*). Since we used the same collection of deletions, it is likely that some of them were at least partially compensated. All of these effects that may have been missing in the individual deletions were present in the newly derived monosomic strains. This caveat actually strengthens our claim that the positive epistatic component can be substantial and even offset most of the total damage associated with the combined individual negative effects.

The advent of systems biology has raised hopes that the 'statistical' and 'biological' sides of epistasis can be coherently brought together (*Moore and Williams, 2005*; *Phillips, 2008*). Indeed, a truly functional explanation of the measured growth effects would require an understanding of how the material and energetic expenditures were altered by the same mutations when they were separate or combined. Such a metabolomic interpretation could only be attempted semi-quantitatively and for much simpler systems (*Molenaar et al., 2009*). Regarding proteome and transcriptome, the latter has the advantage of higher accuracy and repeatability, the ability to detect the products of even weakly expressed genes, and, crucially for this study, to provide clear signals of major functional transitions such as fast-slow growth, anabolism-catabolism, presence-absence of stress. The relationship between the transcriptome and the proteome is not straightforward in multicellular organisms, where compensation for the perturbations introduced by aneuploidy is often observed (*Birchler and Veitia, 2021*). In yeast, most of the changes observed in mRNA are generally reflected as corresponding shifts in both the profile of translation and the composition of mature proteins (*Larrimore et al., 2020*; *Pavelka et al., 2010*). The relationship is not perfect as, for example, disomics have a somewhat attenuated average expression of proteins encoded on the duplicated chromosome. But even there, the result is still much closer to doubling than to parity (*Dephoure et al., 2014*). Post-transcriptional attenuation can be more frequent and pronounced when only single genes are doubled, but this is a very different case from ours (*Ascencio et al., 2021*). Crucially, while signs of compensation at the proteome level are found in wild aneuploids, they are much less pronounced in the laboratory strains and especially in the BY used here (*Messner et al., 2023*; *Muenzner et al., 2024*). (It is tempting to suggest that the wild aneuploids represent a biased sample of aneuploid mutants, i.e., they either already possessed means of compensating for the distortions introduced by aneuploidy or were able to evolve them and thus avoid purging by selection.) In summary, although mRNA analysis does not provide a complete description of the metabolism of the laboratory-generated monosomic cells, it does provide valuable information about it.

The simultaneous upregulation of genes encoding ribosomal proteins (RP) and downregulation of those encoding subunits of the proteasome was observed in all monosomic strains, providing a crucial insight into the functional response of the cell to monosomy. The absence of a single chromosome meant that several genes encoding the translational machinery that resided on it were halved in

**Table 1.** Yeast Slim GO Biological Process categories of the tested deletions and the predicted and observed relative doubling rate of the monosomic strains.

| Monosomic strain | M1 | M2 | M3 | M8 | M10 | M11 | M14 | M16 |
|---|---|---|---|---|---|---|---|---|
| *Subsets of deletions mapping to chromosomes* | | | | | | | | |
| Number of GO Slim Biological Process categories | 25 | 47 | 11 | 27 | 39 | 33 | 34 | 57 |
| Number of unique genes | 6 | 40 | 11 | 20 | 31 | 19 | 24 | 41 |
| | | | | | | | | |
| *Number of genes affecting categories constituting PSA (protein synthesis apparatus):* | | | | | | | | |
| rRNA processing | 1 | 5 | 3 | 7 | 4 | 5 | 8 | 11 |
| Ribosome large subunit biogenesis | | 1 | 1 | | | 4 | 6 | 7 |
| Ribosome small subunit biogenesis | 1 | 4 | 2 | 4 | 3 | 1 | 2 | 5 |
| Ribosomal assembly | 1 | 1 | 1 | 1 | | 2 | 1 | 2 |
| Ribosome subunit export from nucleus | | | | 1 | 2 | 3 | 2 | 4 |
| Cytoplasmic translation | 1 | 9 | | 5 | 4 | 5 | 3 | 5 |
| Translational initiation | 1 | | | | 2 | | | |
| Regulation of translation | | 2 | 2 | | | | | |
| Total number of unique PSA genes | 1 | 13 | 5 | 8 | 10 | 7 | 12 | 14 |
| | | | | | | | | |
| *Relative doubling rate (rDR)* | | | | | | | | |
| Predicted from the total load of deletions (*Figure 2B*) | 0.85 | –0.95 | 0.66 | 0.27 | –0.82 | –0.33 | –0.04 | –1.52 |
| Predicted from the PSA deletions | 0.95 | 0.37 | 0.78 | 0.63 | 0.22 | 0.49 | 0.22 | –0.07 |
| Observed in monosomic strains | 0.96 | 0.60 | 0.63 | 0.61 | 0.49 | 0.60 | 0.63 | 0.58 |

dosage, while dozens of others remained unaffected. The cell could not restore the required stoichiometry by overexpressing the affected genes. Favorable environmental conditions signaled that translation needed to be intensified, but functioning ribosomes were in short supply, resulting in seemingly indiscriminate overproduction of ribosomal proteins and withholding of degradation of both them and other cytosolic proteins. The response was inadequate and costly, but it should be seen as an attempt at ad hoc rebalancing rather than a prepared (evolved) response. This interpretation, strongly suggested by the transcriptomic data, appears even more plausible when analyzing the functional diversity of the 468 genes qualified as haploinsufficient. *Table 1* shows that the diversity was ample: of the 100 Slim GO Biological Process categories, dozens could be linked to each chromosome-associated subset of deletions. However, there was a significant common motif. Processes involved in the 'protein synthesis apparatus' (our term) were most frequently represented. Their predicted damage was also particularly high, typically sufficient or exceeding that required to reduce the growth rate to the levels actually observed in monosomics. The biosynthetic perturbation was thus real, and its relationship to the observed transcriptomic response was likely causal, not merely coincidental.

Some of the other adaptations seen in the monosomal transcriptomes could also be related to the dominant effect of translational inefficiency. A signal to increase biosynthesis would mobilize fermentation and reduce oxidation, which would downregulate genes encoding ROS-scavenging proteins. The Mns2-mediated and other stress responses were apparently absent, contrary to previous reports (*Sheltzer et al., 2012*). The hallmarks of stress responses include decreased RP production and increased proteolysis (to remove destabilized proteins). Here, these two processes were driven in opposite directions, blurring the expected patterns. The non-homogeneity of the response in other regulons analyzed – cAMP-PKA, NCR, GAAC – means that functions other than ribosome assembly were severely affected, but differently, depending on the composition of genes made insufficient by the loss of a chromosome (see also *Figure 4—source data 1*). If there were so many different piecemeal responses, why was the uniform positive epistasis observed?

A possible answer begins with the assumption that the cell is an aggregate of multiple functional modules (*Hartwell et al., 1999*). Our monosomic strains had many different modules that were

negatively affected by partially inactivating mutations. These mutations did not interact with each other directly, but rather through the modules to which they belonged. Mutations that affected the most critical module(s) suppressed the negative effect of mutations that affected other, non-limiting modules. The latter were either less damaged or less needed under current conditions. The epiphenomenon of positive epistasis for fitness reflected the fact that not all mutations exerted their negative effects, at least not to the full extent. In our case, this is not just speculation, but rather the simplest and most prudent explanation linking growth rate and transcriptomic data. Our experimental system is special in a way: a microbial cell in which several modules have been compromised but exposed to favorable conditions and thus tested for the single capacity to grow rapidly. Although peculiar, the system is also instructive. Any unicellular organism is typically tested for only a few specific condition-dependent functions defining fitness, which are very different under, for example, growth or starvation. Similarly, the specialized cells of a complex organism are required to support only a few, and different, elements of life. The modularity of the cell helps us understand how so many partial genetic defects can be carried by so many different organisms without drastically reducing their fitness: rarely or never are all the defects really important.

Interactions within pairs of gene knockouts have received much attention (*Costanzo et al., 2010*; *Segrè et al., 2005*; *Szappanos et al., 2011*). Epistasis for fitness arising from interactions between at least several mutations has also been studied but mostly in the context of 'diminished returns' of successive beneficial variants (*Barrick et al., 2009*; *Chou et al., 2011*; *Khan et al., 2011*; *Kryazhimskiy et al., 2014*). It has been proposed that the pattern may arise from interactions of mutations through a 'global' phenotypic interface, such as fitness (*MacLean et al., 2010*; *Perfeito et al., 2014*; *Schoustra et al., 2016*). However, a similar overall effect could also be produced by 'idiosyncratic' interactions without any mediating factors (*Lyons et al., 2020*; *Reddy and Desai, 2021*). Indeed, this has been demonstrated experimentally, at least for a selected and moderately large set of mutations (*Bakerlee et al., 2022*). Finally, negative epistasis of beneficial mutations has been attributed to cellular modularity by postulating that the positive contribution of each functional module to fitness must have its upper bound (*Wei and Zhang, 2019*). Given that the negative epistasis between multiple positive effects can arise from different mechanisms, we do not insist that the explanation of positive epistasis between multiple negative effects we propose is the only possible one. However, it may be particularly applicable when the deleterious mutations are truly numerous and distributed across many cellular subsystems that work as functional modules.

## Materials and methods
### Single-gene deletion strains
### Selection of single-gene deletions with a possible effect of haploinsufficiency
Deutschbauer et al. have assayed a complete collection of heterozygous single-gene deletion strains and identified a total of 184 genes, 98 essential and 86 non-essential, as haploinsufficient for growth in rich medium, YPD (*Deutschbauer et al., 2005*). We included this set of genes in the present study. Using a different technique, our group have previously tested all 1142 essential heterozygous single-gene deletions and 946 non-essential deletions selected as likely not neutral for growth (*Marek and Korona, 2016*). We reviewed the latter study and, using a false discovery rate of 0.15, accepted up to 404 genes, 256 essential and 148 non-essential, as potentially haploinsufficient. The two sets obtained in two different studies, 184 and 404, overlapped in 112 cases. The overlap was much higher than expected, 14.3, if the two sets were just random samples from among 5200 yeast genes. On the other hand, it was limited and suggested that new growth assays were desirable. The new assays are described below; they included all the unique strains identified in the two studies minus 5 that were not present in our current strain collections and another 3 that were dropped as superfluous for the orthogonal design of experimental blocks described below. (They were also the three least promising based on the earlier assays.) *Figure 2—source data 1* lists the strains.

### Control for single-deletion strains
To correctly quantify the predictably small negative growth effects introduced by heterozygous single-deletion strains, an unbiased and accurate estimate of a wild-type phenotype is required. In the

present study, we attempted to achieve this by using multiple strains as controls rather than a single strain. The reason for this was that the gene deletions used here were constructed over several years by several laboratories and then treated with repeated rounds of propagation, which could lead to genetic divergence. We felt it was risky to use one strain as a control for all the others. We looked for a group of strains that were indistinguishable in terms of growth rate, suggesting that they had not acquired genetic changes that affected this trait. In the case of non-essential genes, we searched the *Saccharomyces* Genome Database and found 25 ORFs originally labeled 'dubious', which are currently almost certainly spurious ORFs, located between other ORFs and no closer than 100 bp to their START or STOP codons. Heterozygous strains carrying deletions of these genes were tested for doubling rate in a manner specific to this study (described below). After repeated assays, 16 deletion strains that were closest to the medium growth performance and not statistically different from each other were selected as controls. Knowing that the non-essential and essential strains differ somewhat in their origin and subsequent handling (*Brachmann et al., 1998*), we decided to derive a separate set of control strains for the latter. We selected 32 essential genes that were in the very center of a single, strong, and narrow modal peak of the frequency distribution of estimates collected for the entire collection of heterozygous essentials (*Marek and Korona, 2016*). Again, after repeated tests of growth rate, 16 strains that were closest to the median and not different from each other were selected to serve as the final control for essential gene deletions. Control strains from both groups are listed in *Figure 2—source data 1*.

## Monosomic strains

### Parents of monosomic strains

In an earlier study, we used 32 strains that had a counter-selectable marker (*URA3*) near the center of each of the 32 chromosome arms and a drug resistance marker (*kan*) near a centromere (*Tutaj et al., 2022*). Pairs of strains with the same centromere marker and the counter-selectable markers located on either the left or right arm of the same chromosome were mated with each other. The resulting diploids were sporulated and tetrads dissected to get triple-marked haploid genotypes, *URA3-kan-URA3*. The latter were mated with a standard BY haploid strain of the opposite mating type. In the final set of 16 diploid strains, each strain had one chromosome triple-marked while its homolog and the remaining 15 chromosome pairs were isogenic with the diploid strain BY4743.

### Derivation and verification of monosomic strains

The diploid strains with triple-marked chromosomes were grown overnight in synthetic complete medium (SC) and then 50–500 µL samples of the resulting cultures were plated on standard 5-fluoroorotic acid (5-FOA5-FOA) plates. Emerging colonies were transferred to new 5-FOA plates and YPD plates with 200 mg/mL geneticin. The goal was to identify variants that were able to grow on the 5-FOA plates but not on the geneticin plates, indicating that all three marker genes may have been lost along with an entire chromosome. Colonies identified in this way were often of different sizes, suggesting that they were genetically heterogeneous. The next criterion was to find variants that produced colonies that were visibly smaller than those of the parental strain, but rarely, though regularly, produced colonies similar to those of the parental strain. Such strains were grown in replicate small cultures (5–100 µL, depending on the monosomic strain), tested for negligible frequency of cells forming large colonies, and collected in larger samples that allowed isolation of DNA in quantities sufficient for next-generation sequencing. Monosomy was considered confirmed when the number of reads for the entire length of a single and expected chromosome was halved. Although simple, this protocol required multiple attempts for some chromosomes because few colonies tended to appear on the 5-FOA plates, most of them remained resistant to geneticin, and those that passed these two criteria did not show the required reversibility, that is, the tendency to occasionally return to normal growth in an apparent step. Even when all these phenotypic criteria were met, sequencing occasionally revealed genomes other than those of pure monosomic origin, that is, with the whole and only one chromosome removed. Nevertheless, once confirmed, the monosomic strains could have been reliably propagated on rich and synthetic media, including that containing 5-FOA, as long as the recurrent appearance of fast-growing colonies was monitored and counteracted.

## Control for monosomic strains

The monosomic strains were all derived by us and the derivation involved our stock of haploid strains BY4741 and BY4742. These two were then crossed to produce a diploid BY4743 that lacked the *URA3*MX4 and *kan*MX4 cassettes and was used as a control in the growth assays of the monosomic strains.

## Estimation of *DR* and *rDR*

The collection of single-gene deletion strains was arrayed on six flat-bottomed 8 × 12-well titration plates with 150 µL aliquots per well. Within each plate, the first and last wells contained clean YPD medium, rows 3 and 10 contained control strains, and the deletion strains occupied the remainder of the plate. Plates were filled with either essential or non-essential deletions accompanied by the 16 control strains listed above. One plate contained both essential and non-essential deletions along with eight essential and eight non-essential control strains. Plates were inoculated at 1–5% from thawed samples and kept non-agitated at 30°C for 48 hr until they reached approximately similar densities of stationary phase cells. Such conditioned microcultures were used to inoculate plates with fresh YPD at 0.5% and maintained at 30°C with shaking at 1000 rpm. Cultures were analyzed for OD (600 nm) every 0.5 hr using a TECAN Infinity reader. Four independent replicates of measurements were performed, starting with independent conditionings. OD readings were used to calculate *DR* (doubling rate). To obtain the *rDR* (relative doubling rate), each *DR* estimate of an experimental strain was divided by the average *DR* of the control strains present on the same plate. In the case of the monosomic strains, the entire protocol was analogous except that all experimental and control strains were kept in one plate. Cultures of monosomic strains used in this assay were tested to contain less than 1% fast-growing cells at the time of OD measurements.

## Analysis of DNA and RNA

The first step in preparation for both DNA and RNA analysis was to collect samples of monosomic cells that would be nearly free of the rapidly growing revertant cells. Individual monosomic strains were grown as replicate microcultures (5–100 µL) in YPD at 30°C to stationary phase. The latter were serially diluted and plated to test for the appearance of large colonies indicating the appearance of compensatory mutations. The microcultures in which more than 99% of the colonies were typical of a particular monosomic strain were pooled. Such tested stationary cultures of monosomic cells were directly used to extract DNA as template for high-coverage sequencing (PE 150, expected read depth ~80). The resulting reads were mapped along standard yeast chromosome sequences using bowtie2 (*Langmead and Salzberg, 2012*), along standard sequences of yeast chromosomes: Ensembl release 100, *S. cerevisiae* genome R64-1-1. Duplicate reads were marked with MarkDuplicates (Picard Toolkit 2019. Broad Institute, GitHub). Samtools (1.15.1) was used for BAM files sorting, indexing, and coverage analysis (*Danecek et al., 2021*).

For RNA, purity-tested cultures of monosomic cells were transferred to fresh YPD and incubated for 4 hr at 30°C with agitation. Total RNA was then extracted using the RiboPure RNA Purification Kit. Three replicates of the monosomic and control BY4743 strain were prepared in this manner. Library preparation and PE 150 sequencing were performed by Novagene. Approximately 20 million read pairs were generated per sample. Quality control of the reads was performed using fastQC v0.11.9 (*Andrew, 2010*). RNA reads were aligned to the above standard sequence using Hisat2 v2.1.0 (*Kim et al., 2015*). The resulting alignment files were sorted and indexed with samtools (1.9). Transcript quantification was performed using cuffquant/cuffnorm v2.2.1 (*Trapnell et al., 2012*). Gene count data normalization ('TMM' method) and differential expression analysis (exact test) were performed in the EdgeR test (*Robinson et al., 2010*). Heat maps were created with Heatmapper (*Babicki et al., 2016*).

For transcriptome analysis with the addition of the mRNA spike, cultures were conditioned as above. The density of the cultures was estimated by counting the cells in the Burker's chamber. The cells were also photographed under a light microscope and the long and short axes of the cells were measured for about 50 randomly selected cells. Using the formula for ellipsoids and assuming that the two shorter axes are equal, the volumes of individual cells and their sums were calculated. Equal amounts of the mRNA spike (ERCC RNA Spike-In Mix, Thermo Fisher) were added to all samples (equal number of cells) from which RNA was extracted using hot formamide (*Shedlovskiy et al.,*

*2017*), followed by RNAzol purification. Subsequent library preparation, sequencing, and analysis were performed as described above with the following modifications. We used stranded RNAseq libraries (dUTP method, Novogene). The merged yeast (Ensembl, R64-1-1) and ERCC (https://assets.thermofisher.com/TFS-Assets/LSG/manuals/ERCC92.zip) reference fasta and gtf files were used for all steps. Hisat2 (v2.1.) was run with the options --rna-strandness RF and -k 20. Transcripts were counted using FeatureCounts (2.0.6) with the following parameters: -F GTF -O -s 2 -t exon -p -R -T 8 --largestOverlap. The raw counts for ERCC transcripts were multiplied by the adjustment factors (RK_02 - 8.40399; RK_03 - 2.72366, RK_04 - 3.58748, RK_01 - 1) to obtain the same amount of spike-in per given cell volume. The counts were then normalized in edgeR (*Robinson et al., 2010*) using the TMM method. Normalized fragment counts per million were extracted using edgeR's cpm function and used for subsequent visualization.

## Acknowledgements

We thank J Bobula for her experimental assistance. The open-access publication of this article was funded by the program 'Excellence Initiative Research University' at the Faculty of Biology of the Jagiellonian University in Kraków, Poland.

## Additional information

### Funding

| Funder | Grant reference number | Author |
| --- | --- | --- |
| Narodowe Centrum Nauki | 2017/25/B/NZ2/01036 | Ryszard Korona |
| Narodowe Centrum Nauki | 2022/47/B/NZ8/00537 | Katarzyna Tomala |
| Uniwersytet Jagielloński w Krakowie | DS/MND/WB/INoS/10/2018 | Hanna Tutaj |

The funders had no role in study design, data collection and interpretation, or the decision to submit the work for publication.

### Author contributions

Hanna Tutaj, Conceptualization, Data curation, Formal analysis, Validation, Investigation, Visualization, Methodology, Writing – review and editing; Katarzyna Tomala, Formal analysis, Funding acquisition; Adrian Pirog, Marzena Marszałek, Investigation; Ryszard Korona, Formal analysis, Funding acquisition, Conceptualization, Investigation, Writing – original draft, Writing – review and editing

### Author ORCIDs

Hanna Tutaj ⓘD https://orcid.org/0000-0003-1539-5300
Katarzyna Tomala ⓘD https://orcid.org/0000-0002-4474-1658
Adrian Pirog ⓘD https://orcid.org/0000-0001-8858-1118
Marzena Marszałek ⓘD https://orcid.org/0000-0001-9175-8850
Ryszard Korona ⓘD https://orcid.org/0000-0002-4329-5908

Reviewer #1 (Public review): https://doi.org/10.7554/eLife.87455.3.sa1
Reviewer #2 (Public review): https://doi.org/10.7554/eLife.87455.3.sa2
Reviewer #3 (Public review): https://doi.org/10.7554/eLife.87455.3.sa3
Author response https://doi.org/10.7554/eLife.87455.3.sa4

## Additional files

### Supplementary files
• MDAR checklist

## Data availability

The whole-genome sequencing reads have been submitted to the NCBI under the BioProject accession number PRJNA895333. The RNA-seq data have been deposited in the GEO database under the accession numbers GSE217944 and GSE276940.

The following datasets were generated:

| Author(s) | Year | Dataset title | Dataset URL | Database and Identifier |
|---|---|---|---|---|
| Tutaj H, Tomala K, Pirog A, Marszałek M, Korona R | 2024 | Whole genome sequencing of *Saccharomyces cerevisiae* monosomic strains | https://www.ncbi.nlm. nih.gov/bioproject/? term=PRJNA895333 | NCBI BioProject, PRJNA895333 |
| Tutaj H, Tomala K, Pirog A, Marszałek M, Korona R | 2024 | Extreme positive epistasis for fitness in monosomic yeast strains - part 1 | https://www.ncbi. nlm.nih.gov/geo/ query/acc.cgi?acc= GSE217944 | NCBI Gene Expression Omnibus, GSE217944 |
| Tutaj H, Tomala K, Pirog A, Marszałek M, Korona R | 2024 | Extreme positive epistasis for fitness in monosomic yeast strains - part 2 | https://www.ncbi. nlm.nih.gov/geo/ query/acc.cgi?acc= GSE276940 | NCBI Gene Expression Omnibus, GSE276940 |

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

## Appendix 1

### Transient monosomy of chromosomes VII and XIII

The procedure for isolating monosomics described in 'Materials and methods' was applied in the same way to all strains with marked chromosomes, including both VII and XIII. For both strains, we tended to obtain urine colonies (on 5-FOA agars) that were mostly large (wild-type size), with small ones accounting for a few percent of all isolates. In the search for potential monosomics, we first concentrated on the small colonies. Out of as many as 100 isolates for each parental strain, not a single one met our criteria. Most were kanR, and none of the rest produced small and largely uniform colonies that would tend to revert to wild-type growth. Despite these discouraging signs, two of the most promising candidates were sequenced for each chromosome. These analyses revealed complex polyploidies rather than regular monosomy.

We then examined the dominant large Ura⁻ colonies. They appeared at a mean frequency of 3.1E-06 and 6.3E-06 for chromosomes VII and XIII, respectively. (Number of colonies divided by the number of cells plated on 5-FOA.) For each parental strain, 108 colonies collected from several separately inoculated selective agars were randomly selected for further testing. All of these isolates grew rapidly when reseeded. Only two of the isolates from the strain carrying the marked chromosome VII proved to be kanR, none in the case of chromosome XIII. To interpret these results, it may be useful to recall that the *URA3* marker was placed in the middle of both the left and right chromosomal arms, while *kan* occupied one of the ORFs closest to a centromere: telomere—*URA3*—*kan*—*URA3*—telomere.

Its homolog was free of all three markers. In principle, two mitotic crossovers between telomere and *URA3* on each arm could produce a progeny cell with two marker-free chromosomes. It would be a regular diploid and thus have a wild-type growth rate. There are two arguments against this proposal. First, double crossovers between *URA3* and *kan* would be similarly frequent, and therefore cells lacking *URA3* but retaining *kan* would not be as rare as observed here. Second, in our previous study (Tutaj et al., 2022), we measured the frequency of crossover on the two chromosomes. Based on our estimates, two simultaneous crossovers located within the distal halves of the arms of chromosomes VII and XIII would occur at a rate of 1.1E-08 and 7.9E-09, respectively, which is about two orders of magnitude lower than observed.

The fast-growing and marker-free isolates were then subjected to sporulation and tetrad dissection. For each chromosome, 8 isolates were randomly selected and at least 10 tetrads were dissected. In each of the 16 strains analyzed, tetrads with four viable haploids formed a clear majority, as expected for the products of meiosis with one regular diploid cell. Two of the eight strains analyzed in this way were further tested by high-coverage DNA sequencing. No sequences of the two markers or the MX4 cassette containing them were found. The three loci in question contained only the expected wild-type sequences with a coverage typical for the rest of the genome, that is, diploid (*Appendix 1—figure 1*).

In summary, we did not find a single slow-growing monosomic strain for chromosomes VII and XIII, despite our intensive search. The regularly occurring fast-growing isolates with the phenotype we were looking for were most unlikely to be the result of recombination events such as a double crossover. They were normal diploids, with the chromosomes in question carrying the phenotypic alleles and DNA sequences characteristic of the unmarked homologs. These results suggest that the marked chromosomes were lost and their unmarked homologs underwent endoreduplication.

### The unsuitability of chromosome XII for monosomy research

The parent strain with the triple-labeled chromosome XII tended to produce unusually numerous colonies on the selective 5-FOA agars, mostly as large as those of the regular diploid strains. We did not analyze these large colonies because we considered them likely to be produced by recombination and thus unrelated to monosomy. This chromosome contains the only yeast rDNA region that is about 1 Mb long and is known to be a strong recombination hotspot. Therefore, the quantitative arguments for excluding double crossovers developed for chromosomes VII and XIII were not applicable here. Small colonies were also more numerous than in other strains. The marker and growth rate reversibility tests were applied to several hundred of them, but failed in almost all cases. A few of the most promising isolates were sequenced and none of them showed the regular monosomy we were looking for. We decided to stop our search for M12 at this point because the presence of rDNA made it impractical.

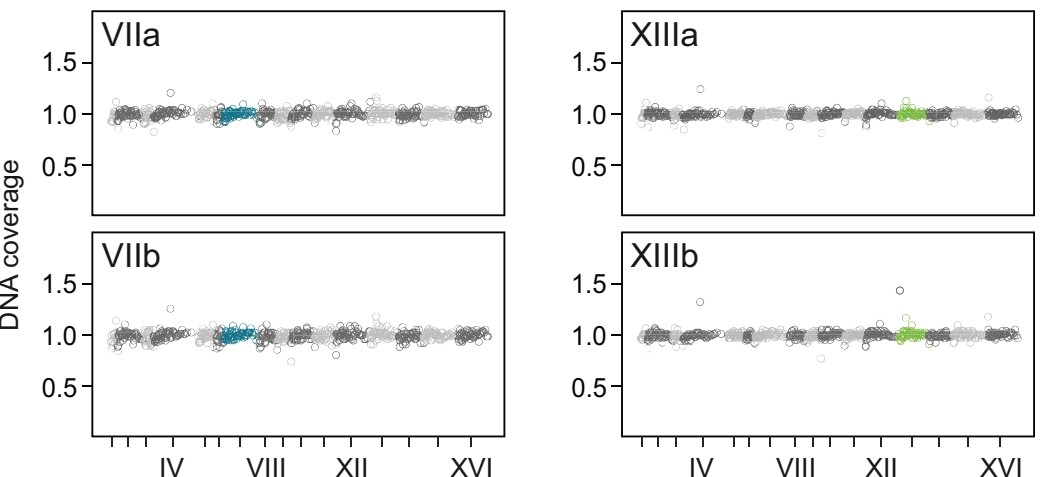

**Appendix 1—figure 1.** DNA whole-genome sequencing coverage after the postulated endoreduplication. Two isolates descending from the parental diploid strains with marked chromosomes VII or XIII are shown. They were subjected to sequencing after being found to lack phenotypic markers and produce four viable spores.

## Appendix 2

### Expected fitness effect of multiple mutations

Fitness is the number of offspring divided by the number of progenitors, $w = No/Np$. This can be the number of cells left by one cell (including itself in the case of budding cells) over a unit of time. Assume that an organism carries multiple mutations—$\alpha$, $\beta$, … $\omega$—which are located in heterozygous loci, their wild-type counterparts are universally marked with +. The fitness effect of a single mutation is $w_{\alpha/+}$, and so on. Fitness can be converted to relative fitness, that is, expressed as a quotient of the wild-type fitness, $w_{\alpha/+}/w_{+/+}$, and so on. Under the multiplicative model of mutation accumulation, an expected joint effect of multiple mutations on relative fitness is a product of individual quotients:

$$w_{exp}/w_{+/+} = \left(w_{\alpha/+}/w_{+/+}\right)\left(w_{\beta/+}/w_{+/+}\right)\ldots\left(w_{w/+}/w_{+/+}\right).$$

When a population is growing continuously, a log transformation of fitness is typically applied because it equals the rate of growth. In particular, it could be the number of doublings completed in a unit of time:

$$\log_2\left(w_{exp}/w_{+/+}\right) = \log_2\left[\left(w_{\alpha/+}/w_{+/+}\right)\left(w_{\beta/+}/w_{+/+}\right)\ldots\left(w_{w/+}/w_{+/+}\right)\right].$$

After replacing the log multiplicative formula above with its log additive equivalent, all terms of the latter can be normalized by dividing by $\log_2$ fitness of the wild-type, which transforms them into relative doubling rates, for example, $rDR_{\alpha/+}=\log_2(w_{\alpha/+})/\log_2(w_{+/+})$. The combined effect of multiple mutations is then equal to

$$rDR_{exp} - 1 = (rDR_{\alpha/+}-1) + (rDR_{\beta/+}-1) + \ldots + (rDR_{\omega/+} - 1)$$

or

$$rDR_{exp} = 1 + \sum d_i$$

where $d = rDR-1$ is an individual mutation ($i$) effect on the relative doubling rate (see **Figure 2B** in the main text).

