## [Editor Report · eLife Assessment]

This study offers **important** insights into the generation and maintenance of monosomic yeast lines and is, to our knowledge, the first to evaluate gene expression in yeast monosomies. The research introduces an innovative method to assess epistasis between genes on the same chromosome, providing **solid** evidence for positive epistatic interactions affecting fitness. Although the authors have substantially improved the methodology and interpretation during revision, questions regarding the interpretation of the transcriptome data have not been completely addressed.

---

## [Referee Report · Reviewer #1 (Public review)]

The study by Korona and colleagues presents a rigorous experimental strategy for generating and maintaining a nearly complete set of monosomic yeast lines, thereby establishing a new standard for studying monosomes. Their careful approach in generating and handling monosome yeast lines, coupled with their use of high-throughput DNA sequencing and RNA sequencing, addresses concerns related to genomic instability and is commendable. However, I would like to express my concerns regarding the second part of the study, particularly the calculation of epistasis and the conclusion that vast positive epistatic effects have been observed. I believe that the conclusion of positive epistasis for fitness might be premature due to potential errors in estimating the expected fitness.

The method used to calculate fitness expectation (1 + sum(di), where di = rDRi - 1) may be inappropriate. The logarithm transformation mentioned by the authors is designed to transform the exponential growth curve into a linear relation for estimating doubling rate, and thus the fitness expectation should be calculated as the product of rDRi values. As an illustration, if gene A exhibits a 20% reduction in fitness when halved (A/-) and gene B exhibits a 30% reduction (B/-), the expected fitness of A/- B/- should be 56%, rather than the 50% estimated in the study. In other words, the formula used by the authors could underestimate the fitness expectation.

This issue is evident in Figure 2b, where negative values were obtained due to the use of an incorrect formula for estimating fitness expectations. It is worth noting that Figure 2a shows rDR values around one, indicating that no further logarithmic transformation was applied.

While widespread positive epistasis in yeast has been reported by other studies (e.g., doi: 10.1038/ng.524, but not to the extent reported in this study), the conclusion of the current study might not be sufficiently supported. I recommend that the authors revisit their calculation methods to provide a more convincing conclusion on the presence of positive epistasis for fitness in their dataset. Overall, I appreciate the authors' efforts in this study but believe that addressing these concerns is essential for strengthening the validity of their findings.

Comments on revised version:

The authors have adequately addressed all my previous concerns during revision.

---

## [Referee Report · Reviewer #2 (Public review)]

This study examines monosomies in yeast in comparison to synthetic lethals resulting from combinations of heterozygous gene deletions that individually have a detrimental effect. The survival of monosomies, albeit with detrimental growth defects, is interpreted as positive epistasis for fitness. Gene expression was examined in monosomies in an attempt to gain insight into why monosomies can survive when multiple heterozygous deletions on the respective chromosome do not. In the RNAseq experiments, many genes were interpreted to be increased in expression and some were interpreted as reduced. Those with the apparent strongest increase were the subunits of the ribosome and those with the apparent strongest decreases were subunits of the proteasome.

The initiation and interpretation of the results were apparently performed in a vacuum of a century of work on genomic balance. Classical work in the flowering plant Datura and in *Drosophila* found that changes in chromosomal dosage would modulate phenotypes in a dosage sensitive manner (for references see Birchler and Veitia, 2021, Cytogenetics and Genome Research 161: 529-550). In terms of molecular studies, the most common modulation across the genome for monosomies is an upregulation (Guo and Birchler, Science 266: 1999-2002; Shi et al. 2021, The Plant Cell 33: 917-939).

It was also apparently performed in a vacuum of results of evolutionary genomics that indicate the classes of genes for which dosage causes fitness consequences. It was from yeast genomics that it was realized that there is a difference in the fate of duplicate genes that are members of molecular complexes following whole genome duplications (WGD) versus small segmental duplications (SSD) with longer retention times from WGD than other genes and an underrepresentation in small scale duplications (e.g. Papp et al. 2003, Nature 424: 194-197; Hakes et al 2007, Genome Biol 8: R209). This pattern arises from negative fitness consequences of deletion of some but not all members of a complex after WGD or the overexpression of individual subunits after SSD (Defoort et al., 2019 Genome Biol Evol 11: 2292-2305; Shi et al., 2020, Mol Biol Evol 37: 2394-2413). In order for this pattern to occur, there must be a reasonably close relationship between mRNA and the respective protein levels. This pattern of retention and underrepresentation has been found throughout eukaryotes (e.g. Tasdighian et al 2017, Plant Cell 29: 2766-2785) indicating that yeast is not an outlier in its behavior.

In the present yeast study, not only are there apparent increases for ribosomal subunits but also for many genes in the GAAC pathway, the NCR pathway, and Msn2p. The word "apparent" is used because RNAseq studies can only determine relative changes in gene expression (Loven et al., 2012, Cell 151: 476-482). Because aneuploidy can change the transcriptome size in general (Yang et al., 2021, The Plant Cell 33: 1016-1041), it is possible and maybe probable that this occurs in yeast monosomies as well. If there is an increase in the general transcriptome size, then there might not be as much reduction of the proteosome subunits as claimed and the increases might be somewhat less than indicated.

Indeed, the authors claim that there is an increased cell volume in the monosomies. Given that cell volume correlates very well with the total transcriptome size (Loven et al., 2012, Cell 151: 476-482; Sun et al 2020, Current Biol 30: 1217-1230; Swaffer et al., 2023, Cell 186: 5254-5268), it could well be that there is an increased transcriptome size in the monosomies. Thus, the interpretation of the relative changes from RNAseq is compromised.

It should be noted that contrary to the claims of the cited paper of Torres et al 2007 (Science 317: 916-924), a reanalysis of the data indicated that yeast disomies have many modulated genes in trans with downregulated genes being more common (Hou et al, 2018, PNAS 115: E11321-E11330). The claim of Torres et al that there are no global modulations in trans is counter to the knowledge that transcription factors are typically dosage sensitive and have multiple targets across the genome. The inverse effect trend is also true of maize disomies (Yang et al., 2021, The Plant Cell 33: 1016-1041), maize trisomies (Shi et al., 2021), Arabidopsis trisomies (Hou et al. 2018), *Drosophila* trisomies (Sun et al. 2013, PNAS 110: 7383-7388; Sun et al., 2013, PNAS 110: 16514-16519; Zhang et al., 2021, Scientific Reports 11: 19679; Zhang et al., genes 12: 1606) and human trisomies (Zhang et al., 2024, genes 15: 637). Taken as a whole it would seem to suggest that there are many inverse relationships of global gene expression with chromosomal dosage in both yeast disomies and monosomies.

In a similar vein, the authors cite Muenzner et al 2024, Nature 630 149-157 that there is an attenuation of protein levels from aneuploid chromosomes while the mRNA levels correlate with gene dosage. This interpretation also seems to have been made in a vacuum of the evolutionary genomics data noted above and there was no consideration of transcriptome size change in the aneuploids. Also, Muenzner et al make the remarkable suggestion that there is degradation of overproduced proteins from hyperploidy, but for monosomies there is greater degradation of the proteins from the remainder of the genome.

To clarify the claims of this study, it would be informative to produce distributions of the various ratios of individual gene expression in monosomy versus diploid as performed by Hou et al. 2018. This will better express the trends of up and down regulation across the genome and whether there are any genes on the varied chromosome that are dosage compensated. The authors claim in the Abstract that "There is no evidence of increased (compensatory) gene expression on the monosomic chromosomes", but then note after describing the increased cell volume of monosomies that this observation likely signals an increased transcriptome size: "Indeed, one explanation for the observed epistasis for viability could be an ample overproduction of all transcripts, so that even those halved by monosomy are sufficiently abundant". It is not clear to this reviewer what conclusions can be made from this work other than the empirical observation that monosomy does not reflect the cumulative effect of multiple haplo-insufficiencies of individual heterozygous deletions and that there are some RELATIVE changes in gene expression, but it is unclear what the ABSOLUTE PER CELL expression is across the whole genome. Clarifying this issue would be important for understanding the nature of any epistasis and fitness consequences.

---

## [Referee Report · Reviewer #3 (Public review)]

The current study examined 13 monosomic yeast strains that lost different individual chromosomes. By comparing the fitness of monosomic strains and several heterozygous deletion strains, the authors observed strong positive epistasis for fitness. The transcriptomes of monosomic strains indicated that general gene-dose compensation is not the reason for fitness gains. On the other hand, gene expression of ribosomal proteins was up-regulated and proteasome subunit expression was down-regulated in all tested monosomic strains. The authors speculated that overexpression in combination with decreased degradation of the insufficient proteins might explain the positive epistasis observed in monosomic strains. This study investigates an important biological question and has some interesting results. However, I have some reservations about the data interpretations listed below.

(1) In Figure 3b (and line 179), the authors stated that those haploinsufficient genes were not transcribed at elevated rates, but almost half of them are in reddish colors (indicating that the expression is higher than 1-fold). Obviously, many haploinsufficient genes are up-regulated in monosomic strains. What the data really show is that the level of overexpression is not correlated with the fitness effect of the deletion (since all the p values are not significant). The authors need to correct their conclusions.

(2) Why are some monosomic strains removed from the transcriptomics analysis, especially when the chromosome IV and XV strains show very strong positive epistasis? The authors need to provide an explanation here.

(3) The authors stated that diploidy observed in chromosome VII and XIII strains were due to endoreplication after losing the marked chromosomes (lines 97 and 117). Isn't chromosome missegregation an equally possible explanation? Since monosomic cells are generated by chromosome missegregation during mitosis, another chromosome missegregation event may occur to rescue the fitness (or viability) of monosomic cells in these strains.

Comments for the revised version:

The authors have addressed all my previous concerns and I have no further questions.

---

## [Author Response]

The following is the authors’ response to the original reviews.

**Public reviews:**

**Response to Reviewer #1 (Public Review):**

The reviewer is correct that the previous explanation of the fitness calculation could be considered insufficient as it was only briefly described in Results. In the revised manuscript, in the "Supplementary Materials" section and then in "Supplementary Text 1", we provide a full definition of the fitness of strains carrying single or multiple mutations and thus show how epistasis was calculated.

**Response to Reviewer #2 (Public Review):**

In our opinion, the reviewer's comments relate to three issues. First, our finding that the level of transcription of the monosomic chromosomes is not upregulated was not compared with the results of other studies, including those in other organisms. Indeed, we did not mention that the gene dosage distortions introduced by aneuploidy are frequently and profoundly compensated in multicellular organisms. We cite the suggested broad and recent review paper in the revised manuscript (line 247). We also removed the somewhat provocative sentence: “The relationship between transcriptome and proteome is generally fixed in yeast”. Regarding this organism, both data and opinions remain indeed conflicting when considering the work with many different yeast strains. But the standard laboratory strains stand out as those where dosage compensation is absent or weak. A paper published a year ago states flatly: "... at least in the strain background used here (authors: BY, the same we use), aneuploidies are transmitted to transcriptome and proteome with a minimum of gene-dosage buffering, rendering aneuploidies discoverable by proteomics" (Messner et al. 2023). A more recent paper reports: "In lab-generated aneuploids, some proteins - especially subunits of protein complexes - show reduced expression, but the overall protein levels correspond to the aneuploid gene dosage" (Muenzner et al. 2024). This "reduced expression" was seen in disomics and was achieved by upregulated proteolysis, whereas we have monosomics and downregulated proteolysis. In summary, we cannot back away from our claim that the biases introduced by monosomy were not compensated. (It is not critical to our paper, we could do it and still leave our main claim about extraordinarily high positive epistasis intact). Muezner and colleagues do report compensation, but in "wild" strains. Our explanation would be that the existing yeast aneuploids are not a random sample of aneuploid mutations. In particular, they could be strains, perhaps relatively rare, in which the genetic background was permissive for aneuploidy from the start or allowed rapid evolution toward tolerance of aneuploidy. Strains with rigid gene-mRNA-protein relationships suffer so much that they perish unless they are shielded from selection, as is possible in the laboratory. The reviewer will know better whether this might also apply to multicellular organisms.

The second concern is that we did not sufficiently report "... the trends of up- and downregulation across the genome and whether there are any genes on the varied chromosome that are dosage compensated". We believe we have indeed done this, albeit mostly in a simple graphical fashion. For the whole genomes, Datasheet 2 reports the extent of down- or up-regulation for each gene in each strain and highlights those that are statistically significant. We do not analyze the distributions of these deviations because they are relative. They represent individual gene down- and up-regulations within a monosomic transcriptome compared to the corresponding genes in the diploid transcriptome, with the total size of the transcriptomes set equal. Thus, the downs and ups cancel each other out, the left and right sides of the distribution would be equal in their totals, and we have no meaningful expectations about the possible variation in the shapes of the overall distributions or their opposite sides. As for the "varied chromosome", we show that there were extensive down- and up-regulations on the monosomic chromosomes, even though the mean expression for them was half that of the diploid chromosomes. This can be seen in Figure 3B as blue and red colored bars that are present on each monosomic chromosome and intermingled along its length. The purpose of these graphs is to show that even the genes in which the halving of the dose was most damaging to fitness (most negative values of rDR-1) did not tend to be upregulated on average (both blue and red colors are found among them). We consider this an important and original part of our data.

Finally, the reviewer is concerned that we are only dealing with the relative abundance of mRNA species. He/she suggests that "... an experiment that would clarify the results would be to perform estimates of the total transcriptome size. If the general transcriptome size is indeed increased, the claims of reduced proteosome expression may need to be revised". We followed this advice and extracted transcriptomes from known amounts of yeast cells with known amounts of standard mRNA or "spike" added. We thus seriously considered the reviewer's suggestion, even though it was contrary to our intuition and, we believe, was not confirmed in the additional experiment. The results are reported in the last paragraph of Results and shown in Supplementary Figure S3. Our arguments are listed in that paragraph, so we will not repeat them here.

**Response to Reviewer #3 (Public Review):**

(1) Figure 3b – both its legend and reference to it in the main text are corrected in line with suggestions made by Reviewers #1 and #3.

(2) We had to restrict our mRNA analysis to about a half of strains. We decided for purely random selection. It left M4 outside but nevertheless included M2, M10 or M16 representing the strains with especially high level of epistasis. See msc. lines 161-162.

(3) We agree, and say so in the article, that both the loss and gain of a copy of a chromosome most likely result in errors in mitosis. By "endoreduplication" we mean any event resulting in two chromosomes instead of one, not necessarily additional DNA replication as we now clarify. We also suggest that both loss and endoreduplication occurred in all strains, but in M7 and M13 they happened so close together that we could not isolate the rare monosomic cells from the rapidly spreading revertants (lines 86-91).

**Recommendations for the authors:**

**Reply to Reviewer #1 (Recommendations for The Authors):**

The legend to Fig. 3b is hopefully clearer now.

**Reply to Reviewer #2 (Recommendations for The Authors):**

We understand that these points were raised also in the public review so the answer to the latter is also relevant to the recommendations for authors.

**Reply to Reviewer #3 (Recommendations for The Authors):**

(1) The first sentence of this comment may be based on a misinterpretation of our main argument. We believe that the upregulation of ribosomal protein (RP) coding genes was not helpful, but harmful. It was costly because RPs are a large part of the proteome, but it did not help translation because it did not restore the stoichiometry of RPs. This unproductive investment reduced the rate of remaining metabolism, so that other impairments introduced by halving the doses of other genes were no longer critical, and this made them unobservable at the level of phenotype, i.e., produced epistasis. However, both this Reviewer and Reviewer 2 seem to suggest that an entire translational apparatus may have been expanded, compensating for its reduced efficiency (per transcript). Reviewer 2 suggested an mRNA spike as a standard, and we followed this approach as more accessible to us. (We reiterate our claim of good agreement between mRNAs and proteins in the BY strain, supported by two new important papers, line 256-257). The results are reported in the last paragraph of Results. We believe that they indicate a reduction, not an increase, in the translational apparatus (including its parts encoded on the monosomal chromosomes), so that our explanation of positive epistasis remains unchallenged.

(2) We re-examined the sequences and found that there were heterozygous SNPs in the same gene, *RSP5*, in several strains. One was a loss of a START codon (M3, M4, M6, M8, M9, M10, M14, M16), always the same. The other was a substitution, always the same, in M5, M11 and M15. There were no mutations in this gene in M1 and M2. We tested our stock haploid strains BY4741 and 4742 and found that they were not mutated. However, we also recovered the specific haploid strains used in the final crosses to construct the diploid strains used to obtain monosomics. Some had one of the two mutations, some were clean. All grew normally, the mutants were similar to the wild types, indicating that the fitness effect of the mutations, even in haploids, was at most partial, since the expected severe effects of *RSP5* inactivation were not visible.

Where do the mutations come from? In previous experiments, we subjected some BY strains to severe selection regimes. As we can now surmise, mutations in *RSP5* helped to resist some of them, especially those involving overexpression of selected genes. (We do not summarize here the results of our lengthy review of our notes and the literature leading this explanation to be the most plausible). Unfortunately, we used strains that went through that harsh selection in crosses serving to derive another collection of strains, those used here.

How critical is it? First, the mutations were heterozygous, which further reduced their apparently weak effects. Second, M1 and M2 were free of them. Third, we tried to get clean monosomics, i.e. with type homozygous for *RSP5*. We obtained such strains with monosomy as the only change for M9, M10 and M16. The other three attempts did not yield correct M3, M5 and M6, but complex aneuploids. This is normal, as we explain (complain) in Results. We would have to isolate a large number of potential monosomies and then sequence them to show that all exact monosomies can be derived in the absence of mutations in *RSP5*. We believe that after an effort comparable to that required to obtain the first set of monosomics, we would complete it. For financial and organizational reasons, this is not possible at this time. We do not consider it necessary to complete the revision. Note that of the five mutation-free straight monosomics, M2, M10 and M16 are among the most affected and thus have the highest positive epistasis. Yes, the role of point mutations cannot be excluded for other monosomics, although we strongly believe it is unlikely. But we have removed all our previous claims that our monosomies are certainly not supported by other genetic changes. Most importantly, our main claim of positive epistasis in its purely descriptive genetic sense remains unaffected. The main functional argument also holds: the indiscriminate overproduction of unbalanced RP proteins was so costly that inefficiencies introduced in functional modules other than biosynthesis become much less relevant. Thus, the main message of our work does not depend on the thinkable, in our view unlikely, role of mutations in *RSP5*.

We provided this lengthy explanation to show that we cared about the reviewer's comment and tried to deal with it in an honest way. It was a lot of pain and no gain for us, but we are still grateful for the opportunity to re-examine our main claims.

(3) The 16 (non-essential) plus 16 (essential) strains were replicated 3 times each. In preliminary experiments, we tested that they were not statistically different (using one-way ANOVA). We considered these 32 strains to have the same genetic background, and thus we considered the 96 estimates homogeneous, except for being influenced only by environmental variation or random error.

(4) We changed the description of Figure 3b to explain that a particular color shows a range (not its boundary) of log2 fold change (FC) relative to the control.

(5) Corrected.

(6) Corrected.